# Refining the domain architecture model of the replication origin firing factor Treslin/TICRR

Pedro Ferreira[1],*, Luis Sanchez-Pulido[2],*, Anika Marko[1], Chris P Ponting[2], Dominik Boos[1]

**Faithful genome duplication requires appropriately controlled replication origin firing. The metazoan origin firing regulation hub Treslin/TICRR and its yeast orthologue Sld3 share the Sld3-Treslin domain and the adjacent TopBP1/Dpb11 interaction domain. We report a revised domain architecture model of Treslin/TICRR. Protein sequence analyses uncovered a conserved Ku70-homologous β-barrel fold in the Treslin/TICRR middle domain (M domain) and in Sld3. Thus, the Sld3-homologous Treslin/TICRR core comprises its three central domains, M domain, Sld3-Treslin domain, and TopBP1/Dpb11 interaction domain, flanked by non-conserved terminal domains, the CIT (conserved in Treslins) and the C terminus. The CIT includes a von Willebrand factor type A domain. Unexpectedly, MTBP, Treslin/TICRR, and Ku70/80 share the same N-terminal domain architecture, von Willebrand factor type A and Ku70-like β-barrels, suggesting a common ancestry. Binding experiments using mutants and the Sld3–Sld7 dimer structure suggest that the Treslin/Sld3 and MTBP/Sld7 β-barrels engage in homotypic interactions, reminiscent of Ku70-Ku80 dimerization. Cells expressing Treslin/TICRR domain mutants indicate that all Sld3-core domains and the non-conserved terminal domains fulfil important functions during origin firing in human cells. Thus, metazoa-specific and widely conserved molecular processes cooperate during metazoan origin firing.**

## Introduction

Accurate and complete DNA replication guarantees faithful genetic inheritance. It requires complex regulation of replication origin firing to ensure (1) efficient firing to avoid non-replicated gaps, and (2) appropriately controlled firing in space and time to facilitate the metazoan genome replication program and coordinate replication with other chromatin processes like transcription (Berezney et al, 2000; Ryba et al, 2010; Helmrich et al, 2013; Dileep et al, 2015; Petryk et al, 2016; Boos & Ferreira, 2019).

Replication initiation is a two-step process in eukaryotes. The first step, origin licensing, in G1 phase is the formation of pre-replicative complex (pre-RC), the loading of the Mcm2-7 replicative helicase onto double-stranded DNA (Evrin et al, 2009; Remus et al, 2009). In pre-RCs, the Mcm2-7 complex does not have helicase activity to avoid premature DNA unwinding in G1. The second step is origin firing, the conversion of pre-RCs into two bidirectional replisomes. Firing occurs S phase-specifically due to its dependency on the S-phase CDKs (S-CDK) and Dbf4-dependent kinase (DDK), whose activities increase at the G1-S transition. During firing, pre-RCs are first remodelled into pre-initiation complexes (pre-ICs) (Zou & Stillman, 1998; Yeeles et al, 2015; Miyazawa-Onami et al, 2017) that then mature into the active Cdc45-Mcm2-7-GINS-DNA polymerase epsilon (CMGE) helicase (Ilves et al, 2010; Langston et al, 2014; Abid Ali et al, 2017; Douglas et al, 2018). DNA synthesis requires assembly of additional replisome factors and primer synthesis (Yeeles et al, 2017).

The main regulation step of origin firing is pre-IC formation. In yeast, a dimer of Sld3 and Sld7 (orthologues of metazoan Treslin/TICRR and MTBP [Kumagai et al, 2010, 2011; Sanchez-Pulido et al, 2010; Sansam et al, 2010; Boos et al, 2011; Boos et al, 2013; Kumagai & Dunphy, 2017; Köhler et al, 2019], binds pre-RCs dependently on pre-RC phosphorylation by DDK (Heller et al, 2011; Deegan et al, 2016). Sld3 recruits Cdc45 via its central Sld3-Treslin domain (STD) domain (Kamimura et al, 2001; Itou et al, 2014) (Fig 1). Sld3 utilizes its TopBP1/Dpb11 interaction domain (TDIN) region to bind to Dpb11 (TopBP1/Cut5/Mus101 in higher eukaryotes) in an interaction that depends on phosphorylation at two CDK sites in the TDIN (Zegerman & Diffley, 2007; Boos et al, 2011). Dpb11 also binds CDK-phosphorylated Sld2 (RecQL4 in higher eukaryotes). Dpb11 and Sld2 form the pre-loading complex together with GINS and DNA polymerase epsilon (Muramatsu et al, 2010). The resulting intermediate structure is called pre-IC. Then, Sld3, Dpb11, and Sld2 dissociate and the CMGE helicase forms.

In addition to cell cycle kinases, the DNA damage checkpoint also controls origin firing at the pre-IC step. Checkpoint kinase phosphorylation of Sld3 and Dbf4 inhibits pre-IC formation to avoid mutations through replicating damaged templates (Lopez-Mosqueda

[1]Molecular Genetics II, Centre for Medical Biotechnology, University of Duisburg-Essen, Essen, Germany  [2]Medical Research Council Human Genetics Unit, Institute of Genetics and Cancer, University of Edinburgh, Edinburgh, UK

Correspondence: dominik.boos@uni-due.de; pedro.ferreira@uni-due.de
*Pedro Ferreira and Luis Sanchez-Pulido contributed equally to this work.

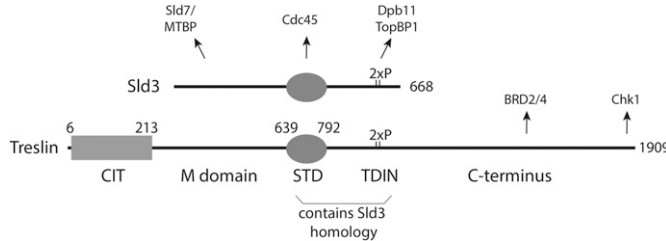

**Figure 1. Treslin/TICRR domain structure.**
CIT, Conserved in Treslins; M, middle domain; STD, Sld3-Treslin domain; TDIN, TopBP1/Dpb11 interaction domain. Numbers indicate amino acid position in human Treslin/TICRR or budding yeast Sld3. Arrows point to interacting proteins: MTBP binds to the Treslin/TICRR M domain, Cdc45 binds to the Sld3-Treslin domain of Sld3 (unknown for Treslin/TICRR), TopBP1 binds to a region containing the two CDK phospho-serine (2xP) residues T969 and S1001 (Boos et al, 2011; Kumagai et al, 2011), Chk1 binds to the very C-terminal 99 amino acids of Treslin (Guo et al, 2015), and Brd2/4 binds to the Treslin/TICRR region 1560–1580 (Sansam et al, 2018).

et al, 2010; Zegerman & Diffley, 2010; Duch et al, 2011). Recently, it has become clear that more subtle regulation of pre-IC factor activity and levels is critical for faithful genome duplication in yeast (Mantiero et al, 2011; Tanaka & Araki, 2011; Tanaka et al, 2011; Reusswig et al, 2016).

Many fundamental processes of yeast origin firing are conserved in vertebrates. All yeast origin firing factors have orthologues in higher eukaryotes (Köhler et al, 2019). In addition, cell cycle regulation by CDK through Treslin/Sld3 binding to TopBP1/Dpb11 and also firing inhibition upon DNA damage through suppression of the Treslin/Sld3-TopBP1/Dpb11 interaction are both conserved (Kumagai et al, 2010, 2011; Sansam et al, 2010; Boos et al, 2011; Guo et al, 2015; Mu et al, 2017).

However, several protein domains of TopBP1, MTBP, Treslin/TICRR, and RecQL4 do not have counterparts in yeasts (Mäkiniemi et al, 2001; Sanchez-Pulido et al, 2010; Zegerman, 2015; Köhler et al, 2019). This suggests that despite the described conservation, metazoa and fungi have evolved specific origin firing processes. Whereas it has been shown that some higher eukaryote-specific domains of MTBP and TopBP1 are required for efficient DNA synthesis (Kumagai et al, 2010; Köhler et al, 2019), the situation for Treslin/TICRR remains less clear. Characterisation of the protein domains that are specific to higher eukaryotes is essential for defining how origin firing processes in these cells diverge from the established yeast model.

The two central STD and TDIN domains of Treslin/TICRR show sequence-based evidence for homology with Sld3 (Fig 1) (Sanchez-Pulido et al, 2010; Boos et al, 2011; Itou et al, 2014). The molecular functions of the STD of Treslin/TICRR and whether this region is essential for replication remain unknown. Its homology with Sld3 suggests that it might support origin firing through interaction with Cdc45 (Itou et al, 2014). The TDIN of Treslin/TICRR is a conserved region containing two CDK phosphorylation sites for TopBP1 binding (Boos et al, 2011; Kumagai et al, 2011). Like the Sld3-TDIN the Treslin/TICRR-TDIN forms a direct binding surface for BRCA1 C-terminal repeat domains (BRCT) in TopBP1/Dpb11 (Zegerman & Diffley, 2007; Boos et al, 2011; Kumagai et al, 2011).

The Treslin/TICRR domains N- and C-terminal of STD and TDIN (Fig 1) have not been shown to be conserved with Sld3. The M

domain shares the ability to bind to MTBP/Sld7 with the N-terminal region of Sld3, and it is required for replication in human cells (Boos et al, 2013; Itou et al, 2015; Köhler et al, 2019). It came as a surprise that sequence conservation with Sld3 was not detected for the Treslin/TICRR M domain, because the interacting regions in MTBP and Sld7, respectively, show homology via remote but statistically significant sequence similarity (Köhler et al, 2019). The C-terminal region of the TDIN is present in many metazoans but is absent from yeast and plants (Sanchez-Pulido et al, 2010). Sequence analysis predicts that this Treslin/TICRR C-terminal region is largely unstructured, with well-conserved stretches of amino acids and more divergent regions alternating. This region binds Chk1 and BRD2/4 (Fig 1), but these activities are not essential for DNA synthesis in cultured human cells (Guo et al, 2015; Sansam et al, 2018). The N-terminal conserved in Treslins (CIT) is conserved in both metazoans and plants, but not present in fungi (Sanchez-Pulido et al, 2010). Whether the CIT functions in replication is unknown.

We here define the essential Sld3-like core of Treslin/TICRR as the three M, STD, and TDIN domains, flanked by higher eukaryote-specific terminal domains. Moreover, we characterise structurally and functionally the M domain and the higher eukaryote-specific terminal regions.

## Results

### The M domain, the STD, and the TDIN domain constitute the essential core of Treslin/TICRR

We first sought to better define the essential core domains of Treslin/TICRR for replication. Mutations of Treslin/TICRR previously showed that the MTBP/Sld7-binding M domain and the TopBP1/Dpb11-binding TDIN perform essential functions during origin firing in human cells (Boos et al, 2011, 2013; Kumagai & Dunphy, 2017). In contrast, the requirement of the Sld3-homologous STD for replication had not previously been addressed in higher eukaryotes. To test this, we used incorporation of the nucleotide analogue BrdU into nascent DNA of cultured human cells in an established RNAi-replacement system (Boos et al, 2011, 2013). U2OS cell clones stably expressing siRNA-resistant Treslin/TICRR WT or STD-deletion mutants (ΔSTD, amino acids 717–792 deleted) to similar levels were treated with control siRNA (siCtr) or Treslin/TICRR siRNA (siTreslin) (Figs 2A and S1A [Blots with siRNA]; Fig S2A–E [data processing strategy]). Cells were pulse-labeled with BrdU 72 h after transfection, stained with anti-BrdU-FITC and propidium iodide (PI), and analysed by flow cytometry. Parental U2OS cells and control cell lines expressing the inactive non-TopBP1 interacting CDK site mutant Treslin/TICRR-2PM showed severely reduced BrdU incorporation levels compared with siCtr-treated cells (Fig 2B). Whereas Treslin/TICRR-WT rescued BrdU incorporation, three independent clones expressing Treslin/TICRR-ΔSTD (clones 11, 17, and 21) showed strong defects in supporting replication (Fig 2B). Quantification of the average replication-dependent BrdU signal in replicates (Fig 2C) (Boos et al, 2013; Köhler et al, 2019; Ferreira et al, 2021) confirmed these observations. Treslin/TICRR-ΔSTD clone 21 rescued replication somewhat better (50% replication) than clones 11 (~30%

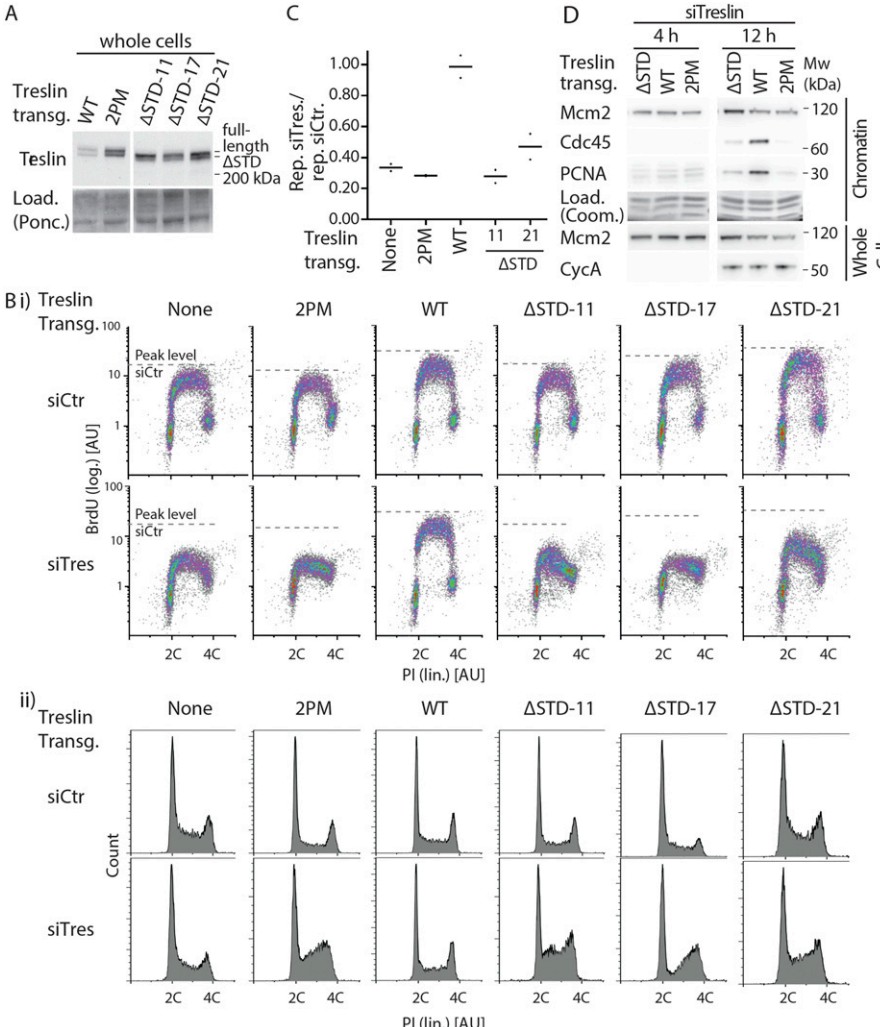

**Figure 2. The Sld3-Treslin domain (STD) domain of Treslin/TICRR is required for DNA replication in cultured human cells.**
**(A)** Whole cell lysates of stable U2OS cell lines carrying siRNA-resistant transgenes of Treslin/TICRR-WT, Treslin/TICRR-2PM (threonine 969 and serine 1001 double alanine mutant that cannot interact with TopBP1 [Boos et al, 2011]), or three clones of Treslin/TICRR with a deletion of the STD (amino acids 717–792 deleted) were immunoblotted with rabbit anti-Treslin/TICRR (amino acids 1566–1909) antibodies. Ponceau (Ponc.) staining controlled for loading (Load.). **(B)** Cells described in (A) were treated with control or Treslin/TICRR siRNAs (siCtr/siTres) before analysis by flow cytometry detecting BrdU (logarithmic [log.] scale) and PI (propidium iodide; linear [lin.] scale). Density plots (i) and PI profiles (ii) are shown. Dashed lines indicating peak level of maximal BrdU incorporation in each cell line upon siCtr-treatment allow visual comparison with level upon siTres treatment. PI profiles histograms show relative cell count. **(C)** Quantification of relative overall DNA replication in cells described in (A) based on flow cytometry experiments described in (B). Averages of BrdU-replication signals of two experiments. Replication signals of siTreslin-treated cells were normalised to replication signals of the same cell line upon siCtr-treatments. **(D)** Stable U2OS cell lines expressing siTreslin-resistant Treslin/TICRR-ΔSTD, WT, or 2PM were released from a double thymidine arrest before treatment with siTreslin and nocodazole. After nocodazole-release for 4 or 12 h cells chromatin was isolated for immunoblotting with goat anti-Mcm2, rat anti-Cdc45, and mouse anti-PCNA antibodies. Whole cell lysates from the same samples were immunoblotted using mouse anti-cyclin A and goat anti-Mcm2 antibodies. For each antibody, crops are from the same immunoblot exposure. Coomassie (Coom.) staining of low molecular weight part including histones controlled for loading. Clone Treslin-ΔSTD -11 was used.

replication; 2PM and no-transgene controls about 30%), exemplifying our observation that individual clones expressing the same transgene showed some variability that probably arise through clonal selection, prompting us to (1) always use more than one clone per mutant throughout the project, and (2) not over-interpret subtle differences between mutants that show less clear defects than Treslin/TICRR-ΔSTD. We then tested if specifically the origin firing step of replication is impaired in Treslin/TICRR-ΔSTD cells by analysing origin licensing and replisome formation on chromatin. Western blotting of chromatin fractions using anti-Mcm2 antibodies showed that replication origin licensing occurred normally in the G1 phase (4 h after Nocodazole release) in Treslin/TICRR-ΔSTD cells. In contrast, origin firing did not occur in the absence of the STD domain as indicated by severely reduced S phase-specific (12 h) Cdc45 and PCNA loading onto chromatin (Fig 2D). The loss of replication activity is not a consequence of a delay in the S-phase entry because cyclin A accumulated normally in Treslin-ΔSTD cells 12 h after release (Fig 2D), and because Treslin/TICRR-ΔSTD cells have a high proportion of S phase cells (Fig 2Bii, PI profiles). We sought to confirm the conclusion that Treslin-ΔSTD cells replicate

slowly because of a defect in origin firing. In an attempt to exclude secondary effects that may complicate interpretation of the presented flow cytometry end point assays (Fig 2B and C), the cells were treated such that the analysed S phase was the first after replacing endogenous Treslin/TICRR with siRNA-resistant transgenes. For this, we monitored cells released from a double thymidine block. A significant fraction of Treslin-WT cells doubled their DNA content within 10 h after release from thymidine, whereas Treslin/TICRR-ΔSTD and Treslin/TICRR-2PM cells accumulated DNA much slower (Fig S3A and B). The fact that any significant DNA synthesis occurred in Treslin/TICRR-ΔSTD, Treslin/TICRR-2PM, and U2OS control cells is likely due to the suboptimal siRNA treatment conditions required in this synchronisation regime (short treatment, only one siRNA round). Immunoblotting chromatin fractions for pre-RC formation (licensing) and replisome formation (firing) revealed that all cell lines contained high levels of pre-RCs in the thymidine arrest (0 h) (Fig S3C and D). In Treslin/TICRR-WT cells, pre-RCs became largely cleared from chromatin 10 h after release, consistent with Mcm proteins being eliminated from chromatin during genome replication through replication termination and

passive replication of origins. In contrast, Treslin/TICRR-ΔSTD, Treslin/TICRR-2PM, and U2OS control cells retained high Mcm2 protein levels 10 h after release, consistent with replication of a large portion of their genome remaining incomplete. Replisomes (PCNA on chromatin) were visible in Treslin/TICRR-WT control cells at early time points, but were severely decreased after 10 h, consistent with genome replication being nearly complete 10 h after release. Lower levels of replisomes also formed in Treslin/TICRR-ΔSTD and Treslin/TICRR-2PM and U2OS control cells due to the inefficient siRNA treatment. However, replisomes were not cleared from the chromatin throughout the entire time course, consistent with slow replication (Fig S3C and D). STD deletion neither led to gross misfolding of Treslin/TICRR nor affected the described activities of the neighbouring M and TDIN domains because Treslin-ΔSTD immunoprecipitated MTBP (Boos et al, 2013) and TopBP1 (Fig S4A and B) normally. Treslin-ΔSTD localised to the nucleus normally (Fig S5). We concluded from these RNAi-rescue experiments that deleting the STD severely compromises replication origin firing in U2OS cells. We concluded from these experiments that, the STD is part of the essential set of core domains of Treslin/TICRR, together with the M and TDIN domains.

### Characterisation of the region N-terminal to the Treslin/TICRR-STD by protein sequence analysis

We then sought to better understand the region N-terminal to the STD of Treslin/TICRR. This has no described sequence conservation with Sld3, but contains the M domain that has a conserved activity—the binding to MTBP/Sld7—and is part of the essential Treslin/TICRR core. To do so we inspected the Treslin/TICRR structure, predicted by Alphafold 2 a recently developed machine learning approach that yields high accuracy (Jumper et al, 2021; Tunyasuvunakool et al, 2021). This predicted structure contained an N-terminal von Willebrand factor type A (vWA) fold (also known as a Rossmann fold, corresponding to the CIT), a β-barrel (corresponding to the M domain, residues 299–424) and the α-helical STD domain (Fig S6A). Unexpectedly, the β-barrel domain was structurally similar to the yeast Ku70 structure (PDB: 5y58_A, residues 264–451; Chen et al, 2018) with a Dali Z-score of 13.3 and a root-mean-square deviation (RMSD) of 2.6 Å (Fig S7). Additional similarities were noted to the known structures of Sld7 (PDB: 3x37_B, residues 3–119; Z-score = 8.6; RMSD = 3.1 Å) and Sld3 (PDB: 3x37_A, residues 4–75; Z-score = 3.1; RMSD = 2.9 Å), thereby identifying Ku70-like β-barrels in both the Sld3 binding domain of yeast Sld7 and the Sld7-binding domain of Sld3 (PDB-ID: 3X37_B) (PDB-ID: 3X37_A) (Figs 3A and S8A–C) (Itou et al, 2015). The Sld3 β-barrel is truncated, containing only five β-strands (Fig S8A). It is notable that the Sld3/Sld7 heterodimer forms in a structurally equivalent manner to the Ku70/Ku80 heterodimer, specifically a homotypic dimer of two structurally similar domains.

Structural similarity could be the result of divergent evolution (i.e., homology) or convergent evolution (i.e., analogy). To distinguish these possibilities, we used iterative profile-to-sequence (HMMer) and profile-to-profile comparisons (HHpred) (Eddy, 1996; Söding et al, 2005; Finn et al, 2011). HHpred searches against the PDB70 profile database (Söding et al, 2005), used the previously identified CIT region that is conserved between animal

and plant Treslins (corresponding to residues 4–254 of human Treslin/TICRR) (Sanchez-Pulido et al, 2010) (Fig 1). This search identified the Treslin/TICRR von vWA domain as homologous to the vWA domain of human complement factor B protein (PDB-ID: 3HRZ_D) (Janssen et al, 2009) ($E$-value = $9.2 \times 10^{-3}$; true positive probability of 97%) (Fig S7). The secondary structure prediction for this region of Treslin/TICRR showed good agreement with the known secondary structure known of diverse members of the vWA superfamily (Jones, 1999) (Fig S7).

In a similar manner, HHpred searches of the Treslin/TICRR M domain against the PDB70 profile database (Söding et al, 2005) yielded statistically significant sequence similarity to yeast Ku70 (PDB-ID: 5Y58_E) (Chen et al, 2018) ($E$-value = 0.3; true positive probability of 88%) (Fig 3A). In further support of homology, the next most statistically significant matches were to three further members of the Ku family, namely yeast Ku80 (PDB-ID: 5Y58_F) (Chen et al, 2018), human XRCC5 (PDB-ID: 1JEY_B), and human XRCC6 (X-ray repair cross-complementing protein 6) (PDB-ID: 1JEY_A) (Walker et al, 2001). Both sequence conservation (HHpred) and Alphafold 2 structure prediction thus provided strong and consistent evidence that the conserved M domain in Treslin/TICRR adopts a Ku70-like β-barrel containing seven core β-strands (Figs 3A and S8). In addition, the structural similarities of the β-barrel domains for Sld3 and Sld7 and their respective human orthologues Treslin/TICRR and MTBP suggest that the Ku70-like β-barrel newly identified in Treslin/TICRR (M domain) is an excellent candidate for being the principal region (heterodimerization domain) that interacts with MTBP.

### The Ku70-like β-barrel of Treslin/TICRR is required for interaction with MTBP

We next tested whether Treslin/TICRR and MTBP may indeed interact via a homotypic Ku70/Ku80-type β-barrel-dependent interaction. Previous biochemical and structural studies had shown that MTBP/Sld7 regions, now established here as part of their β-barrels, interact with Treslin/Sld3 (Itou et al, 2015; Köhler et al, 2019).

We showed previously that deleting two large regions of the Treslin/TICRR M domain, amino acids 265–408 (M1) or 409–593 (M2), compromised MTBP binding (Boos et al, 2013). Deleting M2 abrogated and deleting M1 severely weakened this interaction. Fig 3B shows that a fragment of Treslin/TICRR containing amino acids 260–671 that included M1 and M2 co-immunoprecipitated with endogenous MTBP in lysates of transfected 293T cells. To test the involvement of the Ku70-like β-barrel in Treslin/TICRR, we deleted amino acids 370–400 and 401–420, each containing portions that aligned with Sld3 regions that make direct contacts with Sld7 (Fig 3A, * symbols) (Itou et al, 2015). Both deletions severely compromised the interaction with MTBP (Fig 3B), indicating that the β-barrel is required. To confirm and specify the results from these large deletions, we mutated the three β-strands in the 370–420 region individually (Fig S9A). All strands contain amino acids whose Sld3-equivalents contact Sld7 (Fig 3A, * symbols) (Itou et al, 2015). Seeking to change the amino acid sequence yet preserve the overall structure, we replaced the β-strands by unrelated β-strand forming sequences. Fig S9B shows that all mutations weakened but did not abrogate binding to MTBP. These results are consistent with

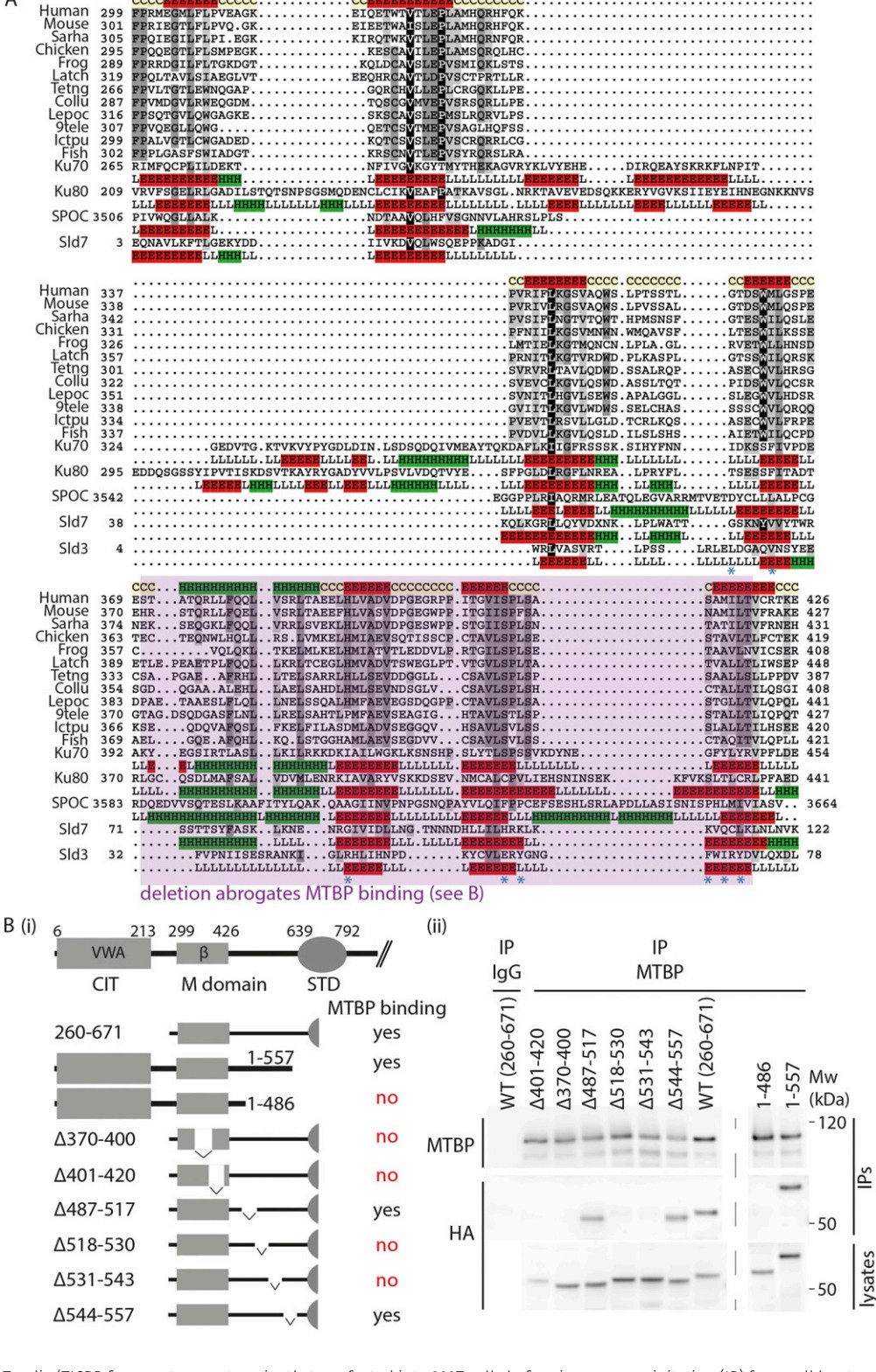

**Figure 3. Treslin/TICRR, Sld3, and Sld7 contain a Ku70/80–like β-barrel that are required for Treslin/Sld3-MTBP/Sld7 dimerization.**

**(A)** Representative multiple sequence alignment of Ku70-like β-barrel domain in the Treslin/TICRR family. The alignment generated with the program T-Coffee (Notredame et al, 2000) using default parameters and slightly refined manually. The final alignment was obtained using a combination of profile-to-profile comparisons (Söding et al, 2005) and sequence alignments derived from structural super-positions of a selection of Ku70-like β-barrel domains whose tertiary structure is known (Holm & Sander, 1995). The limits of the protein sequences included in the alignment are indicated by flanking residue positions. Secondary structure prediction using PsiPred (Jones, 1999) was performed for the Treslin family, shown in the first lane; this prediction is consistent with the secondary structure of Ku70-like β-barrel domains, shown below each of the proteins with known structure (Ku70, PDB: 5Y58E; Ku80, PDB: 5Y58F; SPOC, PDB: 1OW1A; Sld7, PDB: 3X37B; Sld3, PDB: 3X37A). α-helices and β-strands are indicated by H and E, respectively. The alignment was presented with the program Belvu using a colouring scheme indicating the average BLOSUM62 scores (which are correlated with amino acid conservation) of each alignment column: black (>3), grey (between 3 and 1.5) and light grey (between 1.5 and 0.5) (Sonnhammer & Hollich, 2005). Sequences are named according to their specie common name or abbreviation corresponding as follow to their UniProt identification and specie name (Wu et al, 2006): Human, Q7Z2Z1_HUMAN, *Homo sapiens*; Mouse, Q8BQ33_MOUSE, *Mus musculus*; Sarha, G3WMD4_SARHA; *Sarcophilus harrisii*; Chicken, E1BU88_CHICK; *Gallus gallus*; Frog, D3IUT5_XENLA, *Xenopus laevis*; Latch, H3BCK8_LATCH, *Latimeria chalumnae*; Tetng, H3CYF8_TETNG, *Tetraodon nigroviridis*; Collu, A0A4U5UGV6_COLLU, *Collichthys lucidus*; Lepoc, W5ND48_LEPOC, *Lepisosteus oculatus*; 9tele, A0A3B3T1X9_9TELE, *Paramormyrops kingsleyae*; Ictpu, A0A2D0SG01_ICTPU, *Ictalurus punctatus*; Fish, Q6DRL4_DANRE, *Danio rerio*. Blue asterisks: amino acid positions in Sld3 that mediate Sld7 interaction (Itou et al, 2015). **(B)** Schematic representation of Treslin/TICRR mutants (i) used for interaction studies (ii). For (ii), the indicated N-terminally 3HA-tagged Treslin/TICRR fragments were transiently transfected into 293T cells before immunoprecipitation (IP) from cell lysates using control IgG (IgG IP) or rabbit anti-MTBP (amino acids 1–284) (MTBP-IP). Lysates and precipitates were immunoblotted with detection by rat anti-MTBP (12H7) and anti-HA antibodies. VWA, von Willebrand A domain; β, β-barrel.

β-strands in the Treslin/TICRR β-barrel contributing to the MTBP interaction surface that correspond to Sld3/Sld7–interacting strands yet cannot rule out more indirect effects of these mutations.

We found that a region C-terminal to the Ku70-like β-barrel is also required for MTBP interaction. The N-terminal 557 amino acids of Treslin/TICRR, but not the N-terminal 486 amino acids, bound to MTBP (Fig 3B). Small deletions revealed that the amino acids 518–543, but not 487–517 and 545–557, are required for MTBP binding (Fig 3B). The 518-543 region contains a small loop and an α-helical part C-terminal of the β-barrel fold (Fig S6A). In yeast Sld3, a short sequence ~35 amino acids C-terminal to the β-barrel also contains six amino acids that directly contact Sld7 (Itou et al, 2015). We conclude that the Ku70-like β-barrel in the Treslin/TICRR M domain cooperates with a second region further to the C terminus in binding to MTBP. We cannot exclude an indirect contribution of the amino acid 518–543 region to dimerization, although its position in an apparently independent folding unit from the β-barrel makes it unlikely that its deletion destabilised the β-barrel.

Together, our analysis of the N-terminal 600 amino acids of Treslin/TICRR revealed that the structurally conserved part with Sld3 includes the Ku70/80–like β-barrel in the M domain. Thus, the central part of the Treslin/TICRR protein including the M, STD, and TDIN domains constitutes a core that is homologous to Sld3, flanked by Treslin/TICRR-specific terminal domains. Moreover, Treslin/TICRR, MTBP, and Ku70/Ku80 share an N-terminal domain structure comprising a vWA domain followed by Ku70-like β-barrel domains (Fig S6A and B).

## The Sld3-homologous Treslin/TICRR core is insufficient to support replication

We next wanted to test whether the Sld3-like Treslin/TICRR core is sufficient to support replication in human cells or whether it requires the higher eukaryote-specific CIT and C-terminal domains. We performed BrdU-PI flow cytometry upon RNAi-replacement of Treslin/TICRR using mutants that lacked either the CIT (Treslin/TICRR-ΔCIT, amino acids 1–264 deleted), the C-terminal region (Treslin/TICRR-ΔC853, C-terminal 853 amino acids deleted), or both (Treslin/TICRR-core) (Fig 4A). Treslin/TICRR-ΔCIT and Treslin/TICRR-ΔC853 cells showed relatively normal BrdU-PI profiles compared to Treslin/TICRR-WT cells, with S phase populations clearly separated from G1 and G2/M cells by higher BrdU signal intensities (Fig 4B). Quantification of multiple independent experiments indicated only minor reductions in Treslin/TICRR-ΔCIT and Treslin/TICRR-ΔC853 lines (Fig 4C). Testing additional clones confirmed these results (Fig S10A, B, D, and E), although, as described for Treslin/TICRR-ΔSTD, there was some clone-to-clone variability, with one of three ΔC853 clones (no. 29) rescuing like Treslin/TICRR-WT (Fig S10E). Expression levels of Treslin/TICRR-ΔC853 clones were similar or higher than Treslin/TICRR-WT (Figs S1B and S10B and D). The observed clone-to-clone variability makes a clear assessment difficult whether Treslin/TICRR-ΔCIT and ΔC853 are mildly compromised or support DNA replication like Treslin-WT.

Surprisingly, the Treslin/TICRR-core mutant was inactive. BrdU incorporation in Treslin/TICRR-core cells was nearly as strongly

compromised as in the non-replicating control lines (Fig 4B and C, additional clones in Fig S11A–C). This indicated that, albeit individually nonessential for replication, simultaneous deletion of both terminal regions had an additive or even synergistic effect on DNA replication. Treslin/TICRR-core localised normally to the nucleus (Fig S5). Together, the strong reproducible replication defect observed with Treslin/TICRR-core mutants warrants the conclusion that the Sld3-like core domains of Treslin/TICRR require the CIT domain and the C-terminal region to support replication in human cells.

## The CIT cooperates with amino acids 1057–1257 in the C terminus to support origin firing

We then tested which part of the C-terminal region cooperates with the CIT, and whether the cooperation depends on the described binding activities for Chk1 and BRD2/4. We successively truncated the C-terminal sequence in combination with CIT deletion. Neither truncating the Chk1- (Treslin/TICRR-ΔCIT/ΔC99) (Guo et al, 2015) nor the Chk1- and BRD2/4–binding domains (Treslin/TICRR-ΔCIT/ΔC651) (Sansam et al, 2018) recapitulated the synergistic effect (Fig 4A–C; additional clones in Figs S11B, D, and E and S12A–C and E). These double-deletion mutants supported replication to a similar level as Treslin/TICRR-ΔCIT and WT. The C-terminal truncations Treslin/TICRR-ΔC651 and ΔC99 (that contained the CIT) did not greatly affect BrdU incorporation (Figs S10C and S12B). We confirmed these results with two independent double-deletion mutants: Treslin/TICRR-ΔCIT/ΔC309 that contains the BRD2/4 binding site, and Treslin/TICRR-ΔCIT/ΔC394 that does not (Fig S12A, D, and E).

Treslin/TICRR-core did not support replication, as described above. To test whether the known core activities of Treslin/TICRR are intact in the Treslin/TICRR-core protein we tested association with MTBP and TopBP1. Treslin/TICRR-core and ΔC853 co-immunoprecipitated TopBP1 from 293T cell lysates similarly as Treslin/TICRR-ΔC651 (with or without CIT), suggesting that C-terminal deletion of the important amino acids 1057–1257 did not detectably compromise TopBP1 binding (Fig S13A, lanes 4–7 and Fig S13B, lanes 4 and 6). Comparison of Treslin/TICRR-core and ΔC853 with Treslin/TICRR-full-length was difficult because of differences in expression levels and blotting efficiency in transient transfections as a result of considerable size differences. Treslin/TICRR-core also bound MTBP. Some experiments (that had the same limitations as explained for TopBP1 binding experiments) suggested slightly less MTBP bound to Treslin/TICRR-core than to Treslin/TICRR-WT (Figs 4D and S13A and B), which could indicate that the vWA domain-containing CIT makes a small contribution to MTBP binding, similarly to the vWA domain in Ku70/Ku80 (Walker et al, 2001). We cannot formally rule out that potential mild reductions in binding capability of Treslin/TICRR-core to MTBP and TopBP1 fully explains the strong replication deficiency of Treslin/TICRR-core, although this is less likely.

We therefore suggest that two higher eukaryote-specific Treslin/TICRR regions (specifically, CIT and the C-terminal amino acids 1057–1257) have important functions in replication.

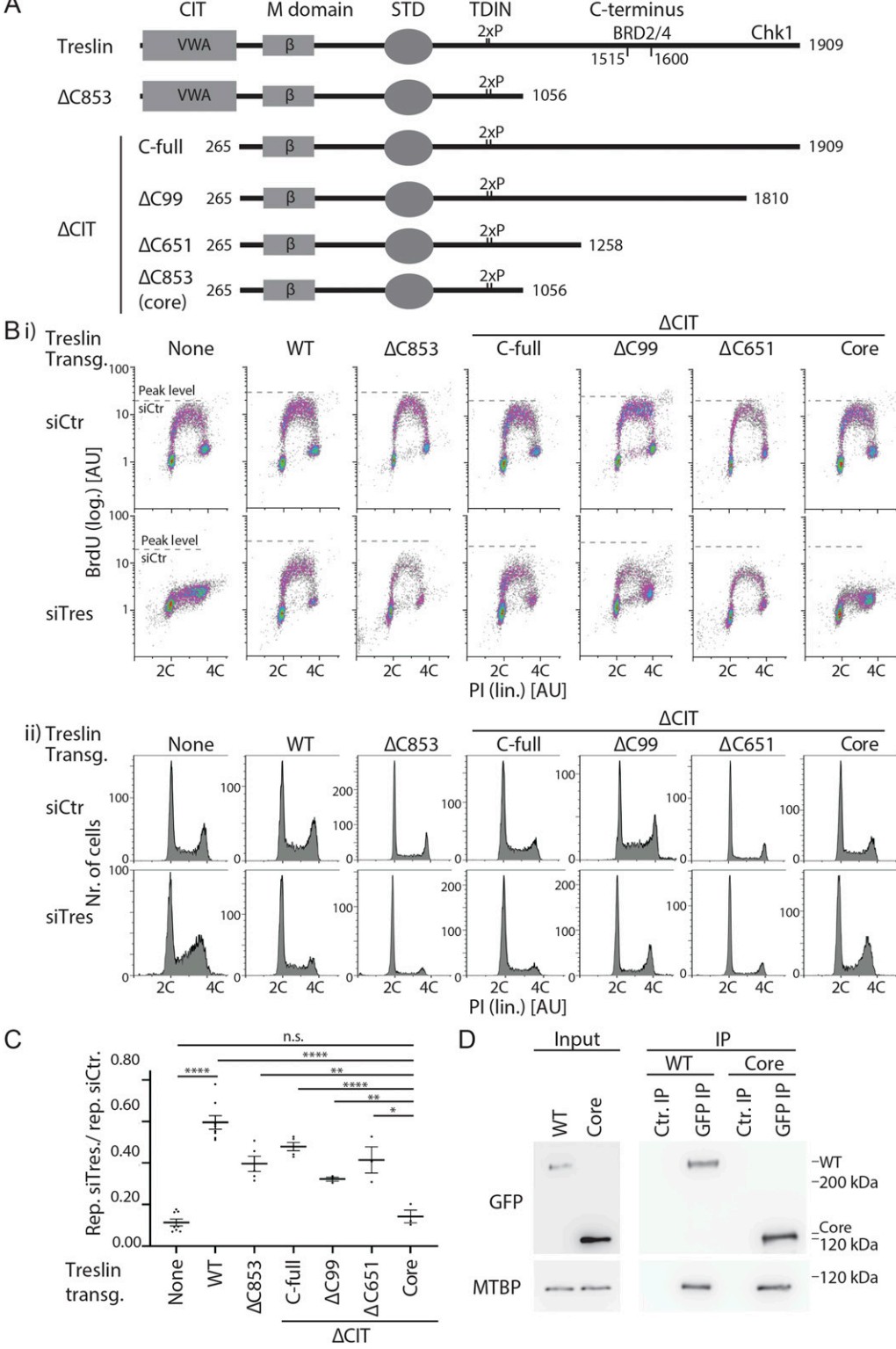

**Figure 4. The conserved in Treslins and the region between amino acids 1057–1257 of Treslin/TICRR cooperate to support replication in human cells.**
**(A)** Schematic representation of Treslin/TICRR mutants used in this figure. Δ, deletion; C99, 651, 853: C-terminal 99, 651, or 853 amino acids, Chk1 kinase binding requires the C-terminal 99 amino acids, BRD2/4 binds to a region between amino acids 1515 and 1600 that were deleted in Treslin/TICRR-ΔC651, -ΔC853, -ΔC394, and -ΔC309 (latter two mutants shown in Fig S5), respectively. ΔCIT, amino acids 1–264 deleted. **(B)** Flow cytometry density plots (i) and propidium iodide profiles (ii) of experiments as described in Fig 2B using the stable U2OS cell lines expressing siTreslin-resistant Treslin/TICRR mutants described in (A). Propidium iodide profiles histograms show relative cell count. Cell clones: ΔC853-5, ΔCIT(-C-full)-5; ΔCIT-ΔC99-25; ΔCIT-ΔC651-61; core-35. **(C)** Quantification of relative overall replication as described in Fig 2C of several independent experiments as described in (B). Cell clones as in (B); Error bars: SEM; sample numbers (n): 8 (none; WT), 5 (ΔCIT[-C-full]; ΔC853), 3 (ΔCIT-ΔC99; ΔCIT-ΔC651; core); significance tests: parametric, unpaired, two tailed t test, *P ≤ 0.05. **(D)** Immunoblot with mouse anti-GFP or rat anti-MTBP (12H7) antibodies of co-immunoprecipitation (IP) experiment using 293T cells transiently transfected with GFP-Flag-Treslin/TICRR-WT or core. Native lysates were immunoprecipitated with anti-GFP nanobodies (GFP-IP) or empty control beads (Ctr. IP).

## Treslin/TICRR-core expressing cells are defective in origin firing

Subtle particularities in cell cycle profiles of Treslin/TICRR-core cells suggested that this mutant may have other defects than cells lacking Treslin/TICRR function. For example, a delay in S phase entry in Treslin/TICRR-core cells could explain the occasionally observed decrease of the S phase sub-population (Fig S11C, clone 41). To exclude such secondary effects of long-term siRNA

treatment as much as possible, we next analysed the first S phase after replacing endogenous with transgenic Treslin/TICRR. We tested whether Treslin/TICRR-core cells licensed origins normally and progressed normally into S phase, but showed a defect in origin firing in. To this end, we released Treslin/TICRR-core-expressing cells and U2OS control cells from a thymidine arrest into a nocodazole block and treated them with siRNA such that they completed S phase before siTreslin could take effect. Upon nocodazole wash-out, U2OS cells typically start replicating at around 7 h, so we chose 4 h and for 12 h to analyse BrdU-PI profiles and replisome formation. All cell lines exited from the nocodazole arrest and entered G1 phase, as indicated by two C DNA content at the 4 h time point (Fig 5A). As usual, a subpopulation of cells released from the arrest with a delay. Subpopulations of siCtr-treated U2OS cells and siTreslin-treated Treslin/TICRR-WT cells had started BrdU incorporation 12 h after nocodazole release. The fastest of these replicating cells had duplicated a significant portion of their genome, as judged by PI signals, showing that they had been replicating for several hours. In contrast, siTreslin-treated Treslin/TICRR-core and control cells incorporated BrdU at nearly undetectable levels. We confirmed that Treslin/TICRR-core cells have a severe defect in genome replication using cells released from a double thymidine arrest. Upon release from the arrest, Treslin/TICRR-core cells accumulated DNA much slower that Treslin/TICRR-WT cells, as measured by PI staining (Fig S3A and B). In addition, immunoblotting of chromatin fractions with Mcm2 and PCNA antibodies revealed that Treslin/TICRR-core cells did not clear pre-RCs from chromatin and replisomes were still visible 10 h after thymidine release (Fig S3E and F). We then tested whether Treslin/TICRR-core expressing cells have defects specifically at the origin firing step of DNA replication, but complete origin licensing and G1-S progression normally. For this, we analysed whole cell lysates and chromatin isolated from nocodazole-released cells. The Mcm2-7 helicase loaded normally onto chromatin in siTreslin-treated Treslin/TICRR-core G1 cells (4 h), showing that licensing was intact (Fig 5B). A cyclin A band was detectable in whole cell lysates after 12 h but not after 4 h in all cell lines, suggesting that Treslin/TICRR core progressed normally into S phase (Fig 5C). In contrast, replisomes did not form more efficiently with Treslin/TICRR-core than in cells without transgenic Treslin/TICRR, as indicated by PCNA and Cdc45 signals on chromatin at 12 h in Treslin/TICRR-WT cells but not in Treslin/TICRR-core and control cells (Fig 5B). The very low signals of Cdc45 and PCNA at 12 h may stem from the siRNA not suppressing endogenous Treslin/TICRR to 100%. We conclude that Treslin/TICRR-core is specifically defective in origin firing.

Together, the Treslin/TICRR terminal regions that are specific to higher eukaryotes cooperate in parallel pathways towards an essential function in replication origin firing.

# Discussion

We here present a characterization of a major origin-firing regulator, Treslin/TICRR, based on its domain structure. Our insight that Treslin/TICRR and Sld3 share similarity of the M domain (Treslin/TICRR) and the N terminus (Sld3), respectively, completes the view

that the three central domains of Treslin/TICRR, M-domain, STD, and TDIN, constitute a Sld3-like core that is flanked by two Treslin/TICRR-specific terminal regions, the CIT and the C-terminal region (Fig 6). These terminal regions are required for Treslin/TICRR's role in replication origin firing.

Important molecular activities of the core domains are known. TDIN is essential for replication in Sld3 and Treslin/TICRR through CDK-mediated interaction with Dpb11 and TopBP1, respectively (Tanaka et al, 2007; Zegerman & Diffley, 2007; Boos et al, 2011; Kumagai et al, 2011). The Sld3-STD binds Cdc45 (Itou et al, 2014), an essential component of the replicative CMG helicase. Although the Cdc45-binding activity of the STD has not been investigated in Treslin/TICRR, conservation with Sld3 suggests that this biochemical activity might also be conserved (Itou et al, 2014). Consistently, we show here that the Treslin/TICRR-STD is required for replication origin firing in cultured human cells, confirming that it has retained important replication functions in humans. The M domain of Treslin/TICRR is also essential for replication in human cells and mediates the binding to MTBP (Boos et al, 2013). Itou et al showed that the M domain-equivalent of Sld3 constitutes a direct binding surface for Sld7 (Itou et al, 2015). We reported earlier that the M domain interacting region in MTBP, approximately the N-terminal MTBP half, contains homology to the Sld3-binding N terminus of Sld7 (Köhler et al, 2019). Here we show that the interaction is mediated by Ku70-like $\beta$-barrel domains in Treslin/TICRR/Sld3 and MTBP/Sld7 (Itou et al, 2015; Köhler et al, 2019), suggesting that they form homotypic dimers comprising structurally similar domains, similar to Ku70-Ku80 dimerization (Walker et al, 2001). Uncharacterised important molecular activities might be situated in the regions between the Treslin/TICRR domains with proven homology to Sld3, such as the DDK-dependent binding to the Mcm2-7 helicase shown for a short stretch of amino acids between the STD and TDIN of Sld3 (Deegan et al, 2016).

We found that the Sld3-like core of Treslin/TICRR was insufficient to support replication and origin firing in U2OS cells, whereas individual deletions of the Treslin/TICRR-specific CIT and C terminus had only mild effects, if any (given the uncertainty due to clonal variability), on Treslin/TICRR's ability to support replication. We concluded that the CIT and the C-terminal region cooperate in parallel pathways to promote DNA replication origin firing. The simplest scenario is that CIT and the C-terminal region promote firing through functions in the molecular process of origin firing that have yet to be revealed. However, more indirect scenarios cannot be excluded. Our finding supports the idea of molecular processes and regulations that are specific to higher eukaryotes to facilitate faithful duplication of their extremely complex genomes. Previous publications had shown roles for higher eukaryote-specific protein domains of TopBP1 (Kumagai et al, 2010) and MTBP (Köhler et al, 2019).

The molecular activities underlying the proposed origin firing functions of CIT and the C-terminal region remain unknown. Our mutants combining CIT-deletion and successive C-terminal truncation excluded significant contributions of the described Chk1- and BRD2/4-binding regions of the Treslin/TICRR C terminus (Guo et al, 2015; Sansam et al, 2018). Instead, comparing the Treslin-ΔCIT/Δ853 with Treslin-ΔCIT/Δ651 mutants suggested that the relevant

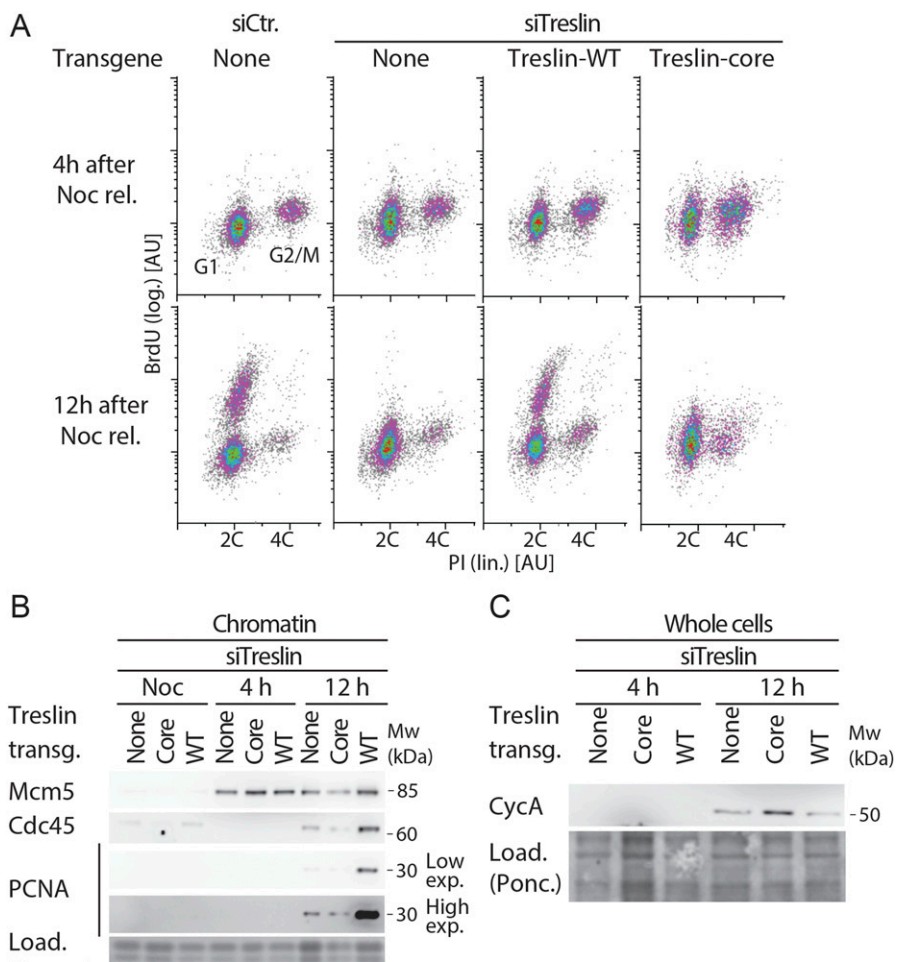

**Figure 5. Treslin/TICRR-core does not support replisome formation.**
**(A)** Stable U2OS cell lines expressing no transgene or siTreslin-resistant Treslin/TICRR-WT or core were released from a thymidine arrest before treatment with siTreslin or siCtr and nocodazole. After nocodazole release for 4 or 12 h cells were analysed by BrdU-propidium iodide flow cytometry. Clone Treslin/TICRR-core-35 was used. **(B)** Chromatin of cells treated as described in (A) was isolated for immunoblotting with rabbit anti-Mcm5, rat anti-Cdc45 and mouse anti-PCNA antibodies. Coomassie (Coom.) staining of low molecular weight part including histones controlled for loading. In the high exposure (exp.) the strongest band is saturated. **(C)** Whole cell lysates of cells treated as described in (A) were immunoblotted using mouse anti-cyclin A antibody.

activity is situated between amino acids 1057 and 1257 of human Treslin/TICRR. Because this region is very close to the TDIN we considered that TopBP1 binding could be compromised in Treslin-Δ853. Although minor defects of Treslin/TICRR-Δ853 mutants in TopBP1 binding cannot be formally excluded we found no clear evidence for a TopBP1 binding deficiency, regardless of whether or not the CIT was present. Also the fact that Treslin-Δ853 mutants that contain the CIT have mild or no defects in supporting genome replication, depending on the clone observed, argues against a significant TopBP1 binding deficiency. A relevant activity in the CIT for origin firing may be to support the binding to MTBP for two reasons: (1) Treslin/TICRR-core and Treslin/TICRR-ΔCIT bound somewhat less well to MTBP (Fig 4D and Köhler et al [2019]), and (2) the CIT-equivalent domain in Ku70/80 makes a small contribution to the Ku70/80 dimer interface (Walker et al, 2001). This potential mild MTBP binding defect may contribute to the inability of Treslin/TICRR-core to support origin firing. However, we find it unlikely that such a moderate defect fully explains the strong replication deficiency of Treslin/TICRR-core. This view is supported by the fact that a Sld3/Sld7-type interaction does not necessarily require a CIT because Sld3 has no CIT domain. We cannot formally exclude that Treslin/TICRR-core is prone to

unfolding, although its normal expression levels, good TopBP1 and MTBP binding capability and normal nuclear localisation speak against this. Other labs also reported that C-terminally deleted Treslin/TICRR-ΔC651 supported replication well (Kumagai et al, 2010), suggesting that C-terminal truncation is compatible with Treslin/TICRR's capability to support replication.

Interestingly, the CIT contains a vWA domain that is also shared by (1) Ku70/Ku80 (Walker et al, 2001) and (2) by MTBP (Fig 6). A specific molecular activity cannot be delineated from the presence of a vWA domain because these domains in other proteins have a variety of activities (Ponting et al, 2000; Whittaker & Hynes, 2002). The Ku70/80 similarities supports speculation that, during evolution, Treslin/TICRR and MTBP received the vWA and β-barrel domains in a single event of genomic recombination. The identical order of the domains in the Ku70/80 proteins suggests that Ku proteins, Treslin/TICRR, and MTBP share an ancestral donor for these domains or that one of the three was the ancestor. Because animal and plant Treslins (but not yeast) contain CITs, the last common ancestor of plants and animals likely contained a CIT. As opisthokonts, fungi and animals are more closely related to each other than animals are to plants, so the CIT must have been lost from Sld3 during yeast evolution. In conclusion, the CIT may have been "donated" to Treslin/

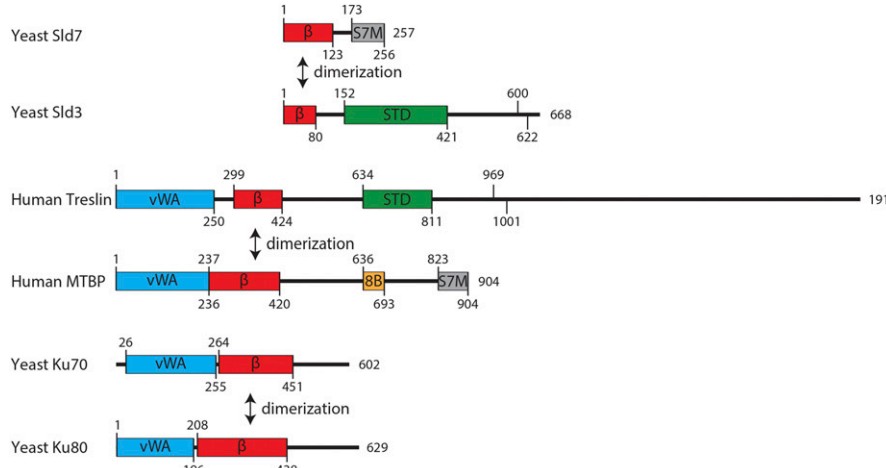

**Figure 6. Common domain architecture of Treslin/TICRR/Sld3, MTBP and Ku70/Ku80 proteins.**
Domain models of the indicated proteins. vWA, von Willebrand factor type A domain; β, Ku70/80–like β-barrel; STD, Sld3-Treslin domain; 8B, Cdk8/19-cyclin C binding domain; S7M, Sld7/MTBP C-terminal domain; Numbers indicate amino acids position and protein length. In Sld3 and Treslin/TICRR are indicated two conserved CDK phosphorylated S/TP sites (Sld3, position 600 and 622; Treslin/TICRR, position 669, 1001).

TICRR as one unit alongside the Ku70-like β-barrel. Both together had the capability to form homotypic dimers with MTBP. The minor (or absent) contribution of the CIT to MTBP binding presents the possibility that it was retained in most branches of evolution because of another function important for eukaryotic cells.

Determining the molecular and cellular functions of the non-core Treslin/TICRR domains will help us better understand the specifics of origin firing in higher eukaryotes compared to yeast. Because Treslin/TICRR mediates origin firing regulation, understanding its non-core domains will likely be necessary to unravel how the complex higher eukaryotic cells coordinate origin firing with other cellular processes.

# Materials and Methods

### Cell culture

U2OS (HTB-96; ATCC) and 293T (CRL-11268; ATCC) cells (both lines kind gift from The Crick institute tissue culture) were cultured in standard conditions in DMEM/high glucose (41965062; Life Technologies), 10% FCS, penicillin/streptomycin in 5% $CO_2$. Stable AcGFP-Flag-GFP-Treslin/TICRR-expressing U2OS cell clones were generated using a pIRES puro3-based vector system by random genome integration followed by selection on 0.3 μg/ml puromycin and picking of individual clones as described (Boos et al, 2011, 2013).

### Analysis of unsynchronised and synchronous stable U2OS cells by BrdU-flow cytometry and chromatin analysis

Endogenous Treslin/TICRR was replaced by siTreslin-resistant transgenes by transfecting U2OS cells twice with Treslin/TICRR siRNA (GAACAAAGGTTATCACAAA) using RNAiMax (13778150; Life Techmologies) as described (Boos et al, 2011). Luciferase siRNA (GL2; Dharmacon) served as a control. For end point analysis of unsynchronised cells, cells were labeled with 10 μM BrdU for 30 min

72 h after the first transfection, harvested and stained with anti-BrdU-FITC (556028; Becton Dickinson) and PI as described (Boos et al, 2011). Flow cytometry analysis was performed, analysed and quantified as described (Köhler et al, 2019). In brief, for quantification of replication rescue using BrdU-PI flow cytometry, the BrdU signal intensity of the S phase cell population was background-subtracted using the combined BrdU-channel signal of G1 and G2/M populations to determine the replication-dependent BrdU signal, as shown in Fig S2A–E. This replication signal was normalised to the replication signal of siCtr-treated cells of the same cell clone to calculate the relative replication rescue. For analysis of synchronized U2OS cells in Fig 2D and E cells were arrested by treatments with 2 mM thymidine for 18 h, release for 10 h, and arrested once again with 2 mM thymidine for 18 h. 4 h after release from the second thymidine block cells were treated with siRNA and 100 μg/ml nocodazole was added for 16 h. Release from the nocodazole arrest was done by washing the cells twice. After cultivation for 4 or 12 h, cells were harvested and analysed by BrdU-flow cytometry as described above or by immunoblotting of whole cell lysates or chromatin-enriched fractions as described (Boos et al, 2013). For Fig 5, cells were instead treated with siRNA and arrested by treatment with 2 mM thymidine for 20 h. Upon release from the thymidine block, 100 μg/ml nocodazole was added for 18 h. Cells were treated with the second round of siRNA 4 h after the start of the nocodazole arrest. For Fig S3 cells were arrested by treatment with 2 mM thymidine for 20 h, released for 10 h, and arrested a second time with 2 mM thymidine for 18 h. Cells were treated with siTreslin or siCtr 8 h after release from the first thymidine arrest. Finally, cells were released from the second thymidine block, harvested 0, 6, or 20 h after release and analysed by PI-flow cytometry or by immunoblotting of chromatin-enriched fractions as described above.

### Antibodies and affinity matrices

Antibodies against Treslin, MTBP, and TopBP1 were described (Boos et al, 2011, 2013; Köhler et al, 2019). Anti-BrdU-FITC (556028; Becton

Dickinson); anti-HA (mouse, 16B12; Covance); anti-GFP nanobodies (kind gift from Kirill Alexandrov); anti-GFP (mouse, JL-8, 632381; Clonetech), anti-Mcm2 (goat, sc-9839; Santa Cruz Biotechnology), anti-Mcm5 (rabbit, ab17967; Abcam), anti-Cdc45 (rat, 3G10; kind gift from Helmut Pospiech), anti-PCNA (mouse, sc-56; Santa Cruz Biotechnology), NHS (N-hydroxysuccinimide) sepharose (10343240; Thermo Fisher Scientific), and Protein G magnetic beads (10004D; Life Technologies).

### Immunoprecipiation from transiently transfected 293T cell lysates

293T cells were transfected using standard calcium phosphate precipitation. 72 h after transfection, cells were harvested and lysed in 5–10 times cell pellet volume using detergent in native lysis buffers and douncing. Lysis buffer for anti-GFP immunoprecipitations in Fig S11 was 20 mM Hepes, 250 mM NaCl, 10% glycerol, 0.1% Triton, 2 mM EDTA, 10 mM NaF, 2 mM mM $\beta$-mercaptoethanol, Complete EDTA-free protease inhibitors (5056489001; Roche); for Fig 4D lysis buffer was 20 mM Hepes, 300 mM NaCl, 10% Glycerol, 0.1% Triton, 2 mM EDTA, 2 mM mM $\beta$-mercaptoethanol, Complete EDTA-free protease inhibitors (5056489001; Roche); for rabbit anti-MTBP immunoprecipitation in Fig 3B 20 mM Hepes, 200 mM NaCl, 10% glycerol, 0.1% Triton, 2 mM mM $\beta$-mercaptoethanol, and Complete EDTA-free protease inhibitors. Lysates from cells from 12.5% (Figs 3B) and 100% (Figs 4D, S3, and S11) confluent 10-cm dish (Figs S3 and S11), as well as 10 $\mu$l (Figs 4D, S3, and S11) GFP nanobody NHS Sepharose beads (1 $\mu$g/$\mu$l) or 1 $\mu$g anti-MTBP (amino acids 1–284) antibody on 10 $\mu$l magnetic protein G slurry beads (Fig 3B) were used per reaction. After washing three times with lysis buffer, beads were boiled in Laemmli loading buffer and analysed by SDS PAGE and immunoblotting. For CDK treatment of lysates, 67 $\mu$g/ml bacterially purified Cdk2-cyclin A (purification system generously donated by Tim Hunt), 5 mM ATP, and 5 mM MgCl$_2$ were added to the lysis buffers.

### Computational protein sequence analysis

Multiple sequence alignments were generated with the program T-Coffee using default parameters (Notredame et al, 2000), slightly refined manually and visualized with the Belvu program (Sonnhammer & Hollich, 2005). Profiles of the alignment as global hidden Markov models were generated using HMMer (Eddy, 1996; Finn et al, 2011). Profile-based sequence searches were performed against the Uniref50 protein sequence database (Wu et al, 2006) using HMMsearch (Eddy, 1996; Finn et al, 2011). Profile-to-profile comparisons were performed using HHpred (Söding et al, 2005). Profile-to-sequence (HMMer) and Profile-to-profile (HHpred) matches were evaluated in terms of an E-value, which is the expected number of non-homologous proteins with a score higher than that obtained for the database match. An E-value much lower than 1 indicates statistical significance. Secondary structure predictions were performed using PsiPred (Jones, 1999). Protein structures and models were analysed using Pymol (http://www.pymol.org). Structure similarity searches and structural superpositions were performed using Dali (Holm, 2020).

## Data Availability

The authors will comply with Life Science Alliance policies for the sharing of research materials and data.

## Supplementary Information

## Acknowledgements

We would like to thank the members of the S Westermann, H Meyer, and D Boos labs for discussion and sharing expertise and regents.

### Author Contributions

P Ferreira: data curation, formal analysis, validation, investigation, visualization, methodology, and writing—original draft.
L Sanchez-Pulido: data curation, formal analysis, validation, investigation, visualization, methodology, and writing—original draft.
A Marko: investigation.
CP Ponting: supervision, validation, and writing—original draft.
D Boos: conceptualization, supervision, funding acquisition, visualization, methodology, writing—original draft, and project administration.

### Conflict of Interest Statement

The authors declare that they have no conflict of interest.

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
