## [Reviewer comments · Life Science Alliance]

Life Science Alliance

Refining the domain architecture model of the replication origin firing factor Treslin/TICRR

Dominik Boos, Pedro Ferreira, Luis Sanchez-Pulido, Anika Marko, and Chris Ponting

DOI: <https://doi.org/10.26508/lsa.202101088>

Corresponding author(s): Dominik Boos, University of Duisburg-Essen and Pedro Ferreira, University of Duisburg-Essen

Review Timeline:	Submission Date:	2021-04-06
	Editorial Decision:	2021-06-25
	Revision Received:	2021-12-03
	Editorial Decision:	2021-12-23
	Revision Received:	2022-01-14
	Accepted:	2022-01-17

Transaction Report:

June 25, 2021

Re: Life Science Alliance manuscript #LSA-2021-01088

Prof. Dominik Boos
University of Duisburg-Essen
Center of Medical Biotechnology, Molecular Genetics II
Universitaetsstrasse 2-5
Essen, NRW, Germany 45138

Dear Dr. Boos,

Thank you for submitting your manuscript entitled "Refining the domain architecture model of the replication origin firing factor Treslin/TICRR" to Life Science Alliance. The manuscript was assessed by expert reviewers, whose comments are appended to this letter. We invite you to submit a revised manuscript addressing the Reviewer comments. Please note that we will not require Reviewer 1's request for X-ray crystallography or NMR. However, all other concerns should be addressed.

The typical timeframe for revisions is three months, however we would be willing to extend this deadline in order for the experimental requests to be satisfied. Please note that papers are generally considered through only one revision cycle, so strong support from the referees on the revised version is needed for acceptance.

Thank you for this interesting contribution to Life Science Alliance. We are looking forward to receiving your revised manuscript.

Sincerely,

B. MANUSCRIPT ORGANIZATION AND FORMATTING:

Reviewer #1 (Comments to the Authors (Required)):

In this work Ferreira et al. extend previous studies of the Treslin protein structure. The authors show that the STD domain of Treslin is necessary for DNA replication in U2OS cells. Through multiple bioinformatic analyses, they then build evidence that the conserved N-terminus is structural similar to vWa domain. They also show that the M domain of Treslin is predicted to fold as a beta-barrel structure, and the structural prediction is conserved in Sld3. Work previously published by the authors showed the importance of two M domains for the binding of Treslin, and the current work showed that smaller deletions in the M1 domain, which contains the beta-barrel, abrogated MTBP binding. The authors then tested whether the core domains of Treslin were sufficient for DNA replication. Although deletion of the N or C-termini outside of the central core domains had little to no effect on DNA synthesis, deletion of both termini disrupted DNA synthesis. The authors further showed that the CIT/VWA in the N-terminus cooperated with C-terminal sequences upstream of those known to interact with Chk1 or Brd4.

Even though the data shown is of good quality, the novel findings in this manuscript are limited, and additional work is needed to support the authors hypotheses. Specifically, supporting the structural predictions with other approaches, such as X-ray crystallography or NMR would greatly enhance the significance of the work. Furthermore, although the identification of essential non-core sequences is intriguing, the significance of the finding is difficult to know without revealing their role at a more mechanistic level. Nonetheless, the descriptive work is of value to the community.

General comments:

1. Throughout the manuscript replicate data from FACS experiments are nicely presented, while only single replicates of co-IP/western blots are shown. For all experiments, data from multiple replicates would help us judge reproducibility.
2. The methods for the structural analyses are insufficiently described. If possible, any code used should be provided. If GUI tools were used, all of the parameters should be described.

Specific comments:

3. In Figure 3, the smaller deletions used to test the involvement of the B-barrel in the MTBP interaction are not substantially informative. If key residues can be predicted based on the structural analysis, testing point mutations in those residues would be more convincing.
4. The second paragraph starting with "Structural comparison with DALI" in the fig S3 legend does not seem to describe anything in the figure.

Reviewer #2 (Comments to the Authors (Required)):

This paper explores the importance of the highly conserved yeast Sld3 "core" of human Treslin/TICCR as well as the CIT and C-terminal domains of human Treslin/TICCR, which plays an important role in origin activation. Therefore, understanding its domain structure is of critical importance.

The main conclusions of this manuscript are that the human CIT and C-terminal domains of Treslin/TICCR do not play a major role in DNA replication and identification of a beta barrel in Treslin that resembles one in Ku70. The authors' findings support current literature that the Sld3 "core" (M, STD and TDIN domains) is highly conserved in Treslin and is needed for its main role in DNA replication. They show this through the use of various Treslin truncation mutants. Additionally, the authors demonstrate that a portion of the C-terminal domain (amino acids 1057-1257) of Treslin plays a role in DNA replication. Overall, this is an important structure-function study of Treslin/TICRR. However, there are several major concerns about the reliance on a single technique, flow cytometry, to demonstrate the effects of truncation mutants on origin firing, the variability in some of the results and missing controls that should be addressed prior to publication. If these issues can be addressed, then I would recommend this study for publication. Specific points of concern are outlined below.

1. As mentioned above, the authors conclusions about specific domains/regions that are necessary for origin firing are based on flow cytometry of BrdU-labeled/PI stained cells. This is a bit concerning as it relies on the proper alignment of BrdU levels based

on the background. Yet, no gating or a BrdU negative control are shown that identify what is counting as BrdU positive cells. Thus, it is hard to properly assess how the "peak levels" were determined. For several samples it is easy to ascertain that replication is severely inhibited but many of the mutants show intermediate levels that are more difficult to assess. It is also not entirely clear what the graphs represent. From the methods, it appears that this is based on BrdU intensity but the histograms all show the "peak levels", which seems somewhat arbitrary. The authors should provide better explanation of how the graphs were generated as well as the gating and BrdU- controls. Histograms of the cell cycle with DAPI only would also be useful. Use of another cell line for key results, while not necessarily required, would also improve confidence in the results. Finally, use of an additional technique such as DNA fiber or chromatin-bound MCM levels across S-phase by flow cytometry or Western blot, or changes in chromatin-bound CMG/PCNA by IF in S-phase cells, are needed for specific mutants to verify the major conclusions of the study.

2. Treslin levels are never shown for the siRNA knockdown. This should be included for each clone along with the levels of the Treslin mutant expression. Perhaps some truncations are not as stable and the effect is more akin to the knockdown as the mutant is may not be expressed well. This appears to be true for some of the mutants shown in the different clones in figures S4 and S7. It is also possible that some of the Treslin truncations do not localize to the nucleus. The truncation mutants all appear to be tagged with GFP, so IF showing they localize to the nucleus should be determine.

3. Fig 5B shows that the Sld3-like core has less MCM and Cdc45 on the chromatin at the 12 hr time point but according to Fig 5A no or little replication has taken place. Can the authors provide an explanation for this?

4. In Fig 1C, a single replicate is shown for clone 17. At least one more experiment should be performed and ideally three replicates should be shown for all samples, like in other figures. In general, the number of independent biological replicates should be stated in the figure legends. This also brings up the issue of variable results across the clones seen in this figure and in figure S4 and S5. As mentioned above, this makes it unclear whether these differences are due to expression, knockdown and/or localization without the proper controls shown (see point 2).

5. In Fig 2A, the Ponceau staining appears uneven and in Fig 2D a loading control is not shown for the whole cell blot. Quantification or a specific loading control should be included.

6. In Fig 2D, the flow cell cycle data for the 4 and 12h release times should be shown, or at least a G1 cell cycle marker, such as Cyclin E or Cdt1, should be included to show that cell cycle profile are similar. Additionally, the "None" control (i.e. wild type levels) are not shown in Fig 2D so one cannot compare what these sample look like compared to "normal". This is important since the "peak levels" differ between clones.

7. In Fig 5C, why is there more Cyclin A in the Core expressing cells compared to the others? Cdt1 or Cyclin E should be included to show that the cells are in S-phase and not arresting at the G1 to S-phase checkpoint.

8. In Fig 1 and 6, the arrows pointing to from Sld3/Treslin to the protein names is not very specific. It would help to show the specific regions of interaction on Sld3/Treslin.

9. Line 56: Treslin/TICRR was also independently discovered in zebrafish by Sansam et al. (Gene Dev. 2010), which should be included in the references here.

10. Line 137 - "proved" is a bit strong given that only two trials are shown with a single technique.

11. Fig 5: The 50% replicated label is confusing. I believe it is to show the mid-point between the 2C and 4C but the purpose of it is not clear, especially since quantification is not included.

Prof. Dr. Dominik Boos and Pedro Ferreira

3rd Dec 2021

Center of Medical Biotechnology
University of Duisburg-Essen
Universitätsstrasse 5, room S05R02H87
45141 Essen, Germany
Phone: +49 201 1834132; mobile: +49 151 42122843
Email: dominik.boos@uni-due.de

Dear reviewers and LSA editors, dear Dr Sawey,

We thank the reviewers and editors for the constructive and efficient review process. We feel that answering the points raised significantly improved the manuscript.

The reviewers were convinced that our study provides valuable insight about the Treslin/TICRR protein and its role in replication origin firing. Reviewer 1's main criticism concerned the insight presented about the structure of the Treslin/TICRR protein. In the revised manuscript we use AlphaFold2 models to support our claims about Treslin/TICRR and MTBP structures, and we generated more specific Treslin/TICRR mutants to test the involvement beta barrel elements in MTBP interaction. Reviewer 2 requested a more clear analysis of flow cytometry data and also independent experimental evidence to strengthen our conclusion about the Treslin/TICRR domain function in genome replication. We provide the requested elements in the revised manuscript.

We also added a new aspect in the revised document. The AlphaFold2 structural models of Treslin/TICRR and MTBP did not only provide independent evidence for the protein sequence analyses presented. The models also allowed the identification of a vWA domain in the N-terminus of MTBP (new Fig 6 and S6). This now shows that all four proteins, Treslin/TICRR, MTBP, Ku70 and Ku80 share a domain architecture in their N-terminal regions, comprising a vWA domain followed by a Ku70 beta barrel, suggesting a common ancestry.

Below we detail how we addressed in the revised manuscript the concerns raised.

Sincerely yours,
Dominik Boos and Pedro Ferreira

Point-by-point response to reviewers' concerns

Reviewer 1:

In this work Ferreira et al. extend previous studies of the Treslin protein structure. The authors show that the STD domain of Treslin is necessary for DNA replication in U2OS cells. Through multiple bioinformatic analyses, they then build evidence that the conserved N-terminus is structural similar to vWa domain. They also show that the M domain of Treslin is predicted to fold as a beta-barrel structure, and the structural prediction is conserved in Sld3. Work previously published by the authors showed the importance of two M domains for the binding of Treslin, and the current work showed that smaller deletions in the M1 domain, which contains the beta-barrel, abrogated MTBP binding. The authors then tested whether the core domains of Treslin were sufficient for DNA replication. Although deletion of the N or C-termini outside of the central core domains had little to no effect on DNA synthesis, deletion of both termini disrupted DNA synthesis. The authors further showed that the CIT/VWA in the N-terminus cooperated with C-terminal sequences upstream of those known to interact with Chk1 or Brd4.

Even though the data shown is of good quality, the novel findings in this manuscript are limited, and additional work is needed to support the authors hypotheses. Specifically, supporting the structural predictions with other approaches, such as X-ray crystallography or NMR would greatly enhance the significance of the work.

We used the Alphafold2 structural models of Treslin/TICRR and MTBP (new Figs. S6 and S8, main text from line 198 and from line 233) to support our original claims based on protein sequence analysis. Alphafold2 predicts vWA and Ku-70-like beta barrels in Treslin/TICRR.

Furthermore, although the identification of essential non-core sequences is intriguing, the significance of the finding is difficult to know without revealing their role at a more mechanistic level. Nonetheless, the descriptive work is of value to the community.

General comments:

1. Throughout the manuscript replicate data from FACS experiments are nicely presented, while only single replicates of co-IP/western blots are shown. For all experiments, data from multiple replicates would help us judge reproducibility.

We thank the reviewer for the trust in our experiments. Regarding non-FACS experiments replicates from key experiments are shown in the revise manuscript in new Fig. S1, new Fig. S4B, and new Fig. S13B.

The new independent experimental approaches provided to functionally test Treslin/TICRR domain mutants lend additional strength to main conclusions (new Fig. S3).

2. *The methods for the structural analyses are insufficiently described. If possible, any code used should be provided. If GUI tools were used, all of the parameters should be described.*

A section was added to the methods part (from line 540) and clarifications added to the main text where appropriate.

Specific comments:

3. *In Figure 3, the smaller deletions used to test the involvement of the B-barrel in the MTBP interaction are not substantially informative. If key residues can be predicted based on the structural analysis, testing point mutations in those residues would be more convincing.*

We provide a new interaction experiment testing more specific mutants of the Ku70-like beta barrel region of Treslin/TICRR. We replaced the individual beta strands in the Treslin/TICRR beta barrel region corresponding to those in Sld3 that make contacts with Sld7. For replacement, we chose sequences that form beta strands but that are not related by sequence in an attempt to preserve the overall beta barrel structure but to change the amino acids. These experiments suggest a contribution of three Treslin/TICRR beta strands to MTBP binding. This evidence is presented in new Fig S9 and described from line 255.

4. *The second paragraph starting with "Structural comparison with DALI" in the fig S3 legend does not seem to describe anything in the figure.*

The Figure and legend have been extended and changed accordingly (new Fig S8).

Reviewer 2:

This paper explores the importance of the highly conserved yeast Sld3 "core" of human Treslin/TICCR as well as the CIT and C-terminal domains of human Treslin/TICCR, which plays an important role in origin activation. Therefore, understanding its domain structure is of critical importance.

The main conclusions of this manuscript are that the human CIT and C-terminal domains of Treslin/TICCR do not play a major role in DNA replication and identification of a beta barrel in Treslin that resembles one in Ku70. The authors' findings support current literature that the Sld3 "core" (M, STD and TDIN domains) is highly conserved in Treslin and is needed for its main role in DNA replication. They show this through the use of various Treslin truncation mutants. Additionally, the authors demonstrate that a portion of the C-terminal domain (amino acids 1057-1257) of Treslin plays a role in DNA replication. Overall, this is an important

structure-function study of Treslin/TICRR.

However, there are several major concerns about the reliance on a single technique, flow cytometry, to demonstrate the effects of truncation mutants on origin firing, the variability in some of the results and missing controls that should be addressed prior to publication. If these issues can be addressed, then I would recommend this study for publication. Specific points of concern are outlined below.

We added more specific assays to confirm the conclusions from the BrdU-flow cytometry experiments (detailed below), and we also provide a better description of how the BrdU-flow cytometry was done, including analysis and quantification.

Our statistics provide prove of reproducibility and statistical significance of the effects measured. We are careful to not draw conclusions based on over-interpreting small differences measured between Treslin/TICRR mutant cell lines that may stem from clonal variability.

1. As mentioned above, the authors conclusions about specific domains/regions that are necessary for origin firing are based on flow cytometry of BrdU-labeled/PI stained cells.

We draw conclusions based on flow cytometry, and in addition based on Western blots detecting on chromatin the relative levels of loaded pre-RCs and replisomes formed.

This is a bit concerning as it relies on the proper alignment of BrdU levels based on the background.

As described below in more detail the background is calculated out and normalised to siCtr cells for quantifications, so that samples are cross-comparable. The provided statistics prove reproducibility between the replicate experiments.

Yet, no gating or a BrdU negative control are shown that identify what is counting as BrdU positive cells. Thus, it is hard to properly assess how the "peak levels" were determined.

We now specify the gating strategy of the flow cytometry and clarify how exactly the analysis and quantification was done (new Fig. S2) to rule out misunderstanding: The replication-dependent BrdU signal is determined relative to BrdU-negative G1 and G2/M populations in the same density dot plot by subtracting the signal of the negative cells (G1/G2/M cells) from the BrdU signal of the S phase cells. The replication-dependent signal in the BrdU channel of siTreslin treated cells is normalised to the replication-dependent BrdU signal in siCtr treated cell of the same cell line in the same experiment, resulting in a value representing % replication relative to control cells that can be cross-compared between cell lines and between experiments. The dashed line labelled "Peak level" provides a visual reference at the

peak of the BrdU arch in siCtr-treated cells of the same experiment in order to compare the effect of siTreslin treatment in the same cell line. We think the line helps, because not all plots shown were taken from the same experiment and therefore have slight differences in Peak-BrdU levels. These differences are calculated out in the described way for quantifications of replication in bar graphs. A no-BrdU control is not necessary because BrdU-negative populations (G1, G2/M) are naturally present in each culture.

For several samples it is easy to ascertain that replication is severely inhibited but many of the mutants show intermediate levels that are more difficult to assess.

We agree, which is why we draw strong conclusions only from flow cytometry data that show strong differences. We have re-formulated this in the main text at a few positions to stress even more the limitations of the method. For example, in line 298 we now state: *“The observed clone-to-clone variability makes a clear assessment difficult whether Treslin/TICRR- Δ CIT and Δ C853 are mildly compromised or support DNA replication like Treslin-WT.”*

It is also not entirely clear what the graphs represent.

We describe this better now in the new Fig S2. The graphs describe averaged replication-dependent BrdU signals relative to siCtr-treated cells of the same line as detailed above. The values of the individual samples used for calculating averages are represented as individual dots. The dot representation makes clear the number of experiments going into the averages, and also give an indication of value scatter between replicates. The significance tests provided indicate the likelihood that the averaged values of two compared samples are truly different.

From the methods, it appears that this is based on BrdU intensity but the histograms all show the “peak levels”, which seems somewhat arbitrary.

The reviewer probably means dot or density plots, not histograms. As explained above, “peak level” lines are just for visual reference of decreased BrdU signal relative to control treated cells. “Peak levels” are not represented in bar graph quantifications.

The authors should provide better explanation of how the graphs were generated as well as the gating and BrdU- controls.

Done, as explained above.

Histograms of the cell cycle with DAPI only would also be useful.

Done, added to the respective Fig 2B, Fig 4B, Fig S10B-D, Fig S11C-D and Fig S12B-D.

Use of another cell line for key results, while not necessarily required, would also

improve confidence in the results. Finally, use of an additional technique such as DNA fiber or chromatin-bound MCM levels across S-phase by flow cytometry or Western blot, or changes in chromatin-bound CMG/PCNA by IF in S-phase cells, are needed for specific mutants to verify the major conclusions of the study.

We now provide experiments using independent techniques to assess replication defects of key Treslin mutants: 1) BrdU pulse and flow cytometry end point assays allow measuring the relative rate of DNA synthesis; 2) Western blotting of chromatin-enriched fractions upon mitotic release into G1 and S phase allows differentiation between defects of mutants in origin licensing (Mcm proteins) and origin firing (replisome generation) (PCNA, Cdc45). Parallel blotting of whole cell lysates for the S phase marker cyclin A rules out defects in S phase entry. Furthermore, we provide new experiments using analysis of synchronised cells in the first S phase after replacing endogenous Treslin/TICRR with the mutants. This allows us to determine replication acutely and to rule out secondary effects by the end point analysis described in 1) and 2); 3) We determine DNA accumulation in cells synchronously progressing through S phase using PI flow cytometry upon release from G1/S block (from line 162 and from line 363). This allows monitoring genome duplication dynamics independently of BrdU. 4) We measure relative amounts of pre-RCs (Mcm proteins) and replisomes on chromatin by western blotting upon release from G1/S block. This measures S phase progression: replisomes form upon origin firing in S phase and exist throughout S phase until replication ends and the last replisomes terminate, whereas pre-RC levels are high after licensing in G1 but are cleared from the DNA by termination and passive replication in the course of S phase.

2. Treslin levels are never shown for the siRNA knockdown. This should be included for each clone along with the levels of the Treslin mutant expression. Perhaps some truncations are not as stable and the effect is more akin to the knockdown as the mutant is may not be expressed well. This appears to be true for some of the mutants shown in the different clones in figures S4 and S7.

Done (New Fig S1).

It is also possible that some of the Treslin truncations do not localize to the nucleus. The truncation mutants all appear to be tagged with GFP, so IF showing they localize to the nucleus should be determine.

We show nuclear localisation of all mutants by IF in new Fig S5 (from line 187 and line 306).

3. Fig 5B shows that the Sld3-like core has less MCM and Cdc45 on the chromatin at the 12 hr time point but according to Fig 5A no or little replication has taken place. Can the authors provide an explanation for this?

Fig 5B shows that, compared to Treslin/TICRR-WT cells, the amount of replisomes (Cdc45 and PCNA) are strongly reduced in cells expressing no transgene and those expressing Treslin/TICRR-core. pre-RC levels are similar in all samples at 4h. Together, this suggests unaffected licensing and strongly reduced replisome formation during origin firing. This goes along with Fig 5A showing strongly reduced BrdU incorporation compared to WT at 12h. 5A cannot exclude that low levels of replication occurs by a small amount of replisomes formed. In fact, very few scattered cells at elevated BrdU signal intensity are visible, suggesting that a very small subpopulation may have escaped siRNA treatment. In addition, G1 populations at 12h of Treslin/TICRR-core and control cells seem to smear up a bit, more so than at 4h, which is consistent with very few replisomes being formed in these cells that result in very low levels of replication. Important is that Treslin/TICRR-core cells are at similarly low replisome levels than siTres-treated control cells, suggesting Treslin/TICRR-core is virtually inactive.

We added a short sentence to acknowledge low, but detectable, BrdU incorporation and replisome levels from lines 362 and 379.

4. In Fig 1C, a single replicate is shown for clone 17. At least one more experiment should be performed and ideally three replicates should be shown for all samples, like in other figures.

Because we felt that the strong effects of all Treslin/TICRR-DSTD clones and the good agreement between replicates (see low scatter between replicates in quantifications) made a third replicate unnecessary. We therefore prioritised other experiments. To acknowledge the low n-number in this experiments we do not provide significance testing in this experiment (we do in all other quantifications). Unfortunately, we lost clone 17. Because we agree that one experiment is not enough for making a quantitative statement, we eliminated this clone from the quantification graph (new Fig 2C).

In general, the number of independent biological replicates should be stated in the figure legends.

We now state the n-number in the legends. We intentionally show quantifications as scatter plots (not bars) to give an impression of the amount of samples used to quantify and of their scatter.

This also brings up the issue of variable results across the clones seen in this figure and in figure S4 and S5. As mentioned above, this makes it unclear whether these differences are due to expression, knockdown and/or localization without the proper controls shown (see point 2).

The quantifications including significance testing of BrdU-FACS experiments prove reproducibility of results with individual clones, meaning that the method is sufficiently accurate to make statistically sound statements. The new experiments

testing the function of mutants (new Fig S3) provide additional confidence. Variability between clones is common when clones are grown from individual HeLa cells that have a high genetic instability. To acknowledge this limitation, we use usually three clones per Treslin/TICRR mutant, and we opt throughout the manuscript to not draw conclusions from small differences between cell lines (e.g. between Treslin/TICRR-WT and deletions of the CIT or the C-terminus). In the revised manuscript we have re-formulated individual sentences to be even more careful to not over-state.

5. In Fig 2A, the Ponceau staining appears uneven and in Fig 2D a loading control is not shown for the whole cell blot. Quantification or a specific loading control should be included.

Considering slight over-loading of DSTD-clone 17 according to Ponceau staining, the blot shown suggests that all Δ STD clones express to levels similar to Treslin/TICRR-WT, with clone 17 expressing a little bit lower than clones 11 and 21. Similar levels of clone 11 and WT are confirmed by our new blot in the new Fig S1. Unfortunately, Fig. 2A is an old blot on X-ray film that cannot be quantified.

6. In Fig 2D, the flow cell cycle data for the 4 and 12h release times should be shown, or at least a G1 cell cycle marker, such as Cyclin E or Cdt1, should be included to show that cell cycle profile are similar.

The PI cell cycle profiles of an equivalent experiment is shown in new Fig 2Bii. It shows that Treslin/TICRR- Δ STD cells show some changes in cell cycle distribution upon replacing endogenous with transgenic Treslin/TICRR, typically slightly increased S and/or G2/M phase populations, like other inactive mutants.

G1 markers are less suitable than S phase markers to monitor cell cycle progression in the Fig 2D experiment, because a subpopulation of cells released from mitosis progress through G1 slowly. This results in a fraction of cells still residing in G1 even when a significant population of cells have progressed significantly through S phase 12h after mitotic release, as seen in Fig 5A, with the same experimental approach. Instead, we used the S phase marker cyclin A to indicate that S phase cells accumulate similarly in all cell lines between 4h and 12 h after release from mitosis. This marker is therefore suitable to show that cell cycle progression delays do not explain the lack of replication in Treslin- Δ STD cells.

In addition, the new experiments shown in the new Fig. S3 provide independent verification. In these experiments the siRNA takes full effect only when cells are arrested at the beginning of S phase by thymidine, which becomes visibly by the fact that siTreslin treated control and Δ STD cells have formed replisomes detectably. However, these cells are subsequently severely compromised in progression of genome replication upon thymidine release.

Additionally, the "None" control (i.e. wild type levels) are not shown in Fig 2D so one cannot compare what these sample look like compared to "normal". This is

important since the "peak levels" differ between clones.

The provided controls siTreslin-treated Treslin/TICRR-WT and Treslin/TICRR-2PM cells provide suitable positive and negative controls, respectively, because these control for the relative levels of pre-RCs and replisomes in replicating and non-replicating cells that were siTreslin-treated, just like Δ STD cells.

7. In Fig 5C, why is there more Cyclin A in the Core expressing cells compared to the others? Cdt1 or Cyclin E should be included to show that the cells are in S-phase and not arresting at the G1 to S-phase checkpoint.

As stated under point 6, Cdt1 or Cyclin E would be problematic to monitor cell cycle progression since a significant fraction of cells in all samples are still in G1. The important point is that Treslin-core cells accumulate cyclin A, so the decrease of replication (FACS) and replisomes (chromatin western) is not due to failure to induce the S phase program.

8. In Fig 1 and 6, the arrows pointing to from Sld3/Treslin to the protein names is not very specific. It would help to show the specific regions of interaction on Sld3/Treslin.

Now specified in the legend of Fig 1.

9. Line 56: Treslin/TICRR was also independently discovered in zebrafish by Sansam et al. (Gene Dev. 2010), which should be included in the references here.

This part of the text specifically refers to papers that proved sequence homology for the first time (Sanchez-Pulido 2010 for Treslin): *"In yeast, a dimer of Sld3 and Sld7 (orthologues of metazoan Treslin/TICRR and MTBP)"*

Sansam et al. 2010 is referenced alongside Kumagai et al. 2010 and others when the text describes Treslin/TICRR function in cells more specifically: *"In addition, cell cycle regulation by CDK through Treslin/Sld3 binding to TopBP1/Dpb11 and also firing inhibition upon DNA damage through suppression of the Treslin/Sld3-TopBP1/Dpb11 interaction are both conserved."* in line 86.

Nevertheless, referencing all these early papers about Treslin and MTBP is justifiable at these position, which is why we have added them.

10. Line 137 - "proved" is a bit strong given that only two trials are shown with a single technique.

Deleted.

11. Fig 5: The 50% replicated label is confusing. I believe it is to show the mid-point between the 2C and 4C but the purpose of it is not clear, especially since quantification is not included.

The line has been taken out of the figure and the text re-formulated: *"The fastest of*

these replicating cells had duplicated a significant portion of their genome, as judged by PI signals, showing that they had been replicating for several hours.” (from line 360).

December 23, 2021

RE: Life Science Alliance Manuscript #LSA-2021-01088R

Prof. Dominik Boos
University of Duisburg-Essen
Center of Medical Biotechnology, Molecular Genetics II
Universitätsstrasse 2-5
Essen, NRW, Germany 45117
Germany

Dear Dr. Boos,

Thank you for submitting your revised manuscript entitled "Refining the domain architecture model of the replication origin firing factor Treslin/TICRR". We would be happy to publish your paper in Life Science Alliance pending final revisions necessary to meet our formatting guidelines. Please address Reviewer 2's final minor points.

- please add ORCID ID for secondary corresponding author-he should have received instructions on how to do so
- please add the Twitter handle of your host institute/organization as well as your own or/and one of the authors in our system
- supplementary references should be incorporated into the main References
- please add callouts for Figures S2A-E; S4A and B; S6B; S8B and C; S11D and E; S12C and S13B to your main manuscript text
- please update the Data Availability section on page 25
- please add molecular weights next to each blot

A. FINAL FILES:

B. MANUSCRIPT ORGANIZATION AND FORMATTING:

Sincerely,

Reviewer #2 (Comments to the Authors (Required)):

The revised manuscript has addressed most of my concerns/comments and the authors have provided adequate explanations for others. The authors are to be commended for their efforts, particularly the inclusion of additional data and explanations that provide transparency to the study. While there is still some concern about the variability of different clones, the authors have allayed these concerns by making clear this issue and not over interpreting the data in the manuscript. The improved methods section, demonstrating how the data was analyzed, addition of key controls and the independent techniques to demonstrate replication defects further improve confidence in their findings and conclusions. Overall, the revised manuscript is significantly improved, and I support its publication. Below are a few minor errors that were found during review:

1. Line 148 and 152: references to clone 17 are still in the text although this data has been removed.
2. Line 175: adding "likely" to the sentence seems appropriate, since there could be other explanations. "...and the U2OS cells is likely due to the suboptimal siRNA treatment..."
3. Line 205: vWA should be defined here in the new text versus in line 225.
4. Line 284: change K70/80 to Ku70/80
5. Line 978: change nucleus to nuclear
6. Supplementary Figure legend S12: the letter and sub-numbering in the figure legend are not correct.
7. Line 1109: add space to C-terminusand

January 17, 2022

RE: Life Science Alliance Manuscript #LSA-2021-01088RR

Prof. Dominik Boos
University of Duisburg-Essen
Center of Medical Biotechnology, Molecular Genetics II
Universitätsstrasse 2-5
Essen, NRW, Germany 45117
Germany

Dear Dr. Boos,

Thank you for submitting your Research Article entitled "Refining the domain architecture model of the replication origin firing factor Treslin/TICRR". It is a pleasure to let you know that your manuscript is now accepted for publication in Life Science Alliance. Congratulations on this interesting work.

DISTRIBUTION OF MATERIALS:

Again, congratulations on a very nice paper. I hope you found the review process to be constructive and are pleased with how the manuscript was handled editorially. We look forward to future exciting submissions from your lab.

Sincerely,
